# *Bacillus pumilus* 15.1, a Strain Active against *Ceratitis capitata*, Contains a Novel Phage and a Phage-Related Particle with Bacteriocin Activity

**DOI:** 10.3390/ijms22158164

**Published:** 2021-07-29

**Authors:** Alberto Fernández-Fernández, Antonio Osuna, Susana Vilchez

**Affiliations:** 1Institute of Biotechnology, Faculty of Sciences, University of Granada, 18071 Granada, Spain; alberto2ff@ugr.es (A.F.-F.); aosuna@ugr.es (A.O.); 2Department of Biochemistry and Molecular Biology I, Faculty of Sciences, University of Granada, 18071 Granada, Spain

**Keywords:** *Bacillus pumilus*, bacteriophage, bacteriocin activity, PBSX, entomopathogen

## Abstract

A 98.1 Kb genomic region from *B. pumilus* 15.1, a strain isolated as an entomopathogen toward *C. capitata*, the Mediterranean fruit fly, has been characterised in search of potential virulence factors. The 98.1 Kb region shows a high number of phage-related protein-coding ORFs. Two regions with different phylogenetic origins, one with 28.7 Kb in size, highly conserved in *Bacillus* strains, and one with 60.2 Kb in size, scarcely found in *Bacillus* genomes are differentiated. The content of each region is thoroughly characterised using comparative studies. This study demonstrates that these two regions are responsible for the production, after mitomycin induction, of a phage-like particle that packages DNA from the host bacterium and a novel phage for *B. pumilus*, respectively. Both the phage-like particles and the novel phage are observed and characterised by TEM, and some of their structural proteins are identified by protein fingerprinting. In addition, it is found that the phage-like particle shows bacteriocin activity toward other *B. pumilus* strains. The effect of the phage-like particles and the phage in the toxicity of the strain toward *C. capitata* is also evaluated.

## 1. Introduction

*Bacillus pumilus* 15.1, a natural bacterium isolated from a partially decomposed common reed plant as a strain active against larvae of the Mediterranean fruit fly (*Ceratitis capitata*) [1], has been extensively characterised in our research group for the last 10 years. *B. pumilus* 15.1 is a motile bacterium, shows the high capability of metabolising a wide range of compounds, and can form biofilms [2]. The strain contains a 7.8 Kb cryptic plasmid, with 33 copies per chromosome [3], and at least one uncharacterised megaplasmid. The strain genome has been previously described [4] and is publicly available (GenBank accession number LBDK00000000.1). *B. pumilus* 15.1 produces, during the sporulation phase, a parasporal crystal [2] resembling the one produced by *Bacillus thuringiensis*, a well-known entomopathogen. The crystal is of proteinaceous nature, is solubilised at low temperature and is mainly composed of the enzyme oxalate decarboxylase. The enzyme can catalyse the transformation of oxalate to formate, a previously recognised neurotoxic compound. It has been proposed that oxalate decarboxylase could be a potential virulence factor [5], although the mechanism of toxicity is still under study.

Many phage and phage-related structures can be associated with virulence factors in bacteria, such as the cholera toxin from *Corynebacterium diphtheriae* [6], the Shiga toxin from the enterohaemorrhagic *Escherichia coli* [7] or the botulism toxin from *Clostridium botulinum* [8,9].

Bacteriophages or phages are viruses that use bacteria as a host, either replicating, causing the death of the host (lytic cycle), or inserting their genome into the bacterial chromosome (prophage) and linking the viral multiplication to the bacterial growth cycle (lysogenic cycle). The ‘decision’ between lysis and lysogeny depends on the phage nature (not all phages are capable of lysogeny), host physiological state, genetic compatibility and host and phage abundance, among other factors [10].

Bacteriophages are extremely common in nature. Early studies showed that most of the bacteria (60–70%) whose genomes have been sequenced have at least one prophage in their genomes [11]. However, recent studies show that this percentage is around 30% [12]. Lysogens can reach up to 90% in some species, such as *Pectobacterium* spp. or *Dickeya* spp. [13]. In the case of *Bacillus* species, such as *B. thuringiensis*, all of the strains investigated for prophage prediction contained putative prophage regions, and 81% of the strains (50 out of 61) have complete prophage regions with an average of 2.21 prophages/genome [14].

Bacteriophages display high levels of mosaicism in their genomes. This level of mosaicism is particularly relevant in Lambdoid phages. It is explained, in part, by the large number of horizontal gene transfer events that take place thanks to the action of recombinases, often encoded by phages themselves [15]. Prophages and the so-called cryptic (or defective) prophages represent the largest reservoir of functional genes for temperate phages [11]. Virulent and defective prophages can exchange genes recovering the activity in the case of defective prophages or modifying the activity of the virulent ones [16].

Some prophages also contain what is known as lysogenic conversion genes [17], independent transcriptional units with non-essential functions for phage replication, but which can change bacterial fitness, enhancing the survival under certain conditions of the lysogenic bacteria over the non-lysogenic one. These lysogenic conversion genes can range from antibiotic resistance genes [18], to the production of toxins [9], or genes that provide increased adhesion properties to the host bacteria [19], among others.

It is widely accepted that once phage DNA is inserted in a bacterial genome, the prophage might be subjected to mutations or partial deletions that could lead to ‘phage domestication’ by losing genes or through gene inactivation, becoming the so-called cryptic or defective phages [20]. This process of domestication had led to the evolution of multiple molecular systems, which are morphologically similar to phage structures, and provide significant advantages to the bacteria that bear them. Among such systems, Gene Transfer Agents (GTAs) [21], Phage-Inducible Chromosomal Islands (PICIs) [22], bacterial type VI secretion systems (T6SS) [23], some bacteriocins, such as pyocins [24], the *Serratia* anti-feeding prophage (AFP) [25] or the *Photorhabdus virulence* cassette (PVC) [26] can be mentioned.

GTAs are phage-like particles with the ability to pack small fragments of host DNA instead of the phage-like particle genome and delivering them into a recipient bacterial cell. The DNA fragments packaged in GTAs particles are smaller than their coding region, and therefore, are incapable of self-replication by horizontal transfer. GTA production is highly controlled by host systems, including quorum-sensing [27] and is restricted to ~3% of the population [28], so the vast majority of the population can be a receptor of the DNA encapsulated in these particles.

PICIs are the Phage Inducible Chromosomal Island family of mobile elements. The best characterised are the *Staphylococcus aureus* Pathogenic Islands (SAPIs), a 15 Kb mobile element that encodes virulence factors and uses helper phages for transmission [29].

Bacteriocins are ribosomally-synthesised peptides produced by bacteria with a narrow bactericide activity against related strains. Among them, the phage-tail-like *Pseudomonas aeruginosa* pyocins are a kind of bacteriocins similar to bacteriophage structures. R-type pyocins are syringe-like contractile particles, similar to phage tails from the Myoviridae family, which insert themselves into the bacterial membrane and kill the bacteria by forming pores in the membrane and disrupting the membrane potential [30]. In addition, F-type pyocins also resemble phage tails, but in this case to the non-contractile form of siphoviridae tails [24].

Although all of the previously described systems are very different in their function (injecting DNA from bacteria to bacteria in the case of GTAs, injecting virulence factors from bacteria to eukaryotic cells in the case of PVCs and AFPs, injecting lytic enzymes from bacteria to bacteria in the case of pyocins, or secreting toxic effectors in the case of T6SS) all have the common characteristic that they are evolutionarily-related phage structures designed to efficiently transport macromolecules from bacteria to other cells.

The PBSX particle, a phage-like particle produced by *Bacillus subtilis*, is another phage-related molecular system that shows some interesting features [31]. This phage-like particle is a mitomycin-inducible structure (dependent on the SOS system) that shows bactericidal activity against *B. subtilis* related bacterial strains, similar to bacteriocins [32]. The PBSX particle is encoded by an approximately 33 Kb region identified in the *B. subtilis* 168 chromosome (Accession number NC_000964.3) [33]. The most remarkable feature of PBSX particles is that they package random 13 Kb DNA fragments from the host cell [34] inside the head when induced with mitomycin.

The interesting functions that phages and phage-related systems represent for bacteria is an ongoing research area in Microbiology and Molecular Biology, especially in the case of pathogenic bacteria, as a high number of the remarkable features that pathogens show are related to them.

With the aim of searching for potential virulence factors in *B. pumilus* 15.1 strain, bioinformatics analysis was performed, and a region of 113 kb in the genome of the bacteria (Contig 59), containing a high number of phage proteins coding ORFs was found. Contig 59 contained a 98.1 Kb *in silico*-predicted prophage present in its sequence. We found that part of the 98.1 Kb region showed a genomic organisation and structure similar to *B. subtilis* PBSX particles. The PBSX-like particle, called Bp15.1PLP, was observed under TEM in mitomycin-induced cultures and was concentrated by polyethylene glycol precipitation. The Bp15.1PLP packages host bacterial DNA in its capsid, with an average size of 9 Kb, instead of the 13 Kb DNA fragments that *B. subtilis* PBSX packages. In addition, *B. pumilus* PBSX-like particles show bactericidal activity against all the *B. pumilus* strains assayed. The major structural proteins of the particle were analysed by SDS-PAGE and identified by MALDI-TOF, and the possible effect of the Bp15.1PLP particle on bacterial toxicity on *C. capitata* larvae was investigated. Apart from Bp15.1PLP, a novel *B. pumilus* phage, named Bp15.1Hope, was also found in the mitomycin-induced *B. pumilus* 15.1 lysates. It was proven that Bp15.1Hope genome is also contained in the studied 98.1 Kb region (Contig 59).

## 2. Results and Discussion

### 2.1. Prophage Prediction in the B. pumilus 15.1 Genome

During characterisation of the strain *B. pumilus* 15.1 [2], we observed that under some culture conditions (such as elevated temperatures (45–50 °C) or oxygen depletion), a phage-like lysis process occurred, as the optical density of the cultures suddenly dropped after a normal initial growth phase (unpublished work). This observation reminded us of the behaviour of the induction of the lytic cycle of a temperate phage and led us to hypothesise the presence of a prophage in the genome of *B. pumilus* 15.1.

The draft genome sequence of *B. pumilus* 15.1, previously published by our group [4] and available under the accession number LBDK00000000.1, was analysed for the detection of prophage regions using PHASTER [35]. Only one region in the bacterial genome, with a potential phage organisation, was identified in contig 59 (NZ_LBDK01000059).

The in silico analysis of the phage-predicted region, with a length of 98.1 Kb (Figure 1), showed that it contained 127 predicted open reading frames (ORFs), with an average of 44.4% in GC content, a GC content higher than the average reported by *B. pumilus* strains (41.6% GC). That was a striking fact, as previous studies have established that bacteriophage usually has lower GC content than their hosts [36,37]. One tRNA for Alanine was also identified with the tRNA-scanSE algorithm with a score of 98.8.

A preliminary comparative analysis of the sequence identified by PHASTER at nucleotide level showed two distinct regions, with different prevalence levels in databases. A region of 28.695 Kb, widely spread in the genome of bacteria belonging to the *B. pumilus* group, was detected at the 5’ end of the 98.1 Kb region (Figure 1; shadowed in green), while another region of 60.213 Kb, at the 3´ end of the 98.1 Kb region (shadowed in purple), contained nucleotide sequences rarely found in sequenced genomes from bacteria belonging to the *B. pumilus* group (only five matches in the NCBI database; specified below). This region was flanked by a repeat sequence of 45 bp that comprised part of the tRNA-Ala identified by tRNAScan-se.

The two identified regions showed a significantly different GC content (Figure 1); while the 28.7 Kb region showed a GC content (40.5%) similar to the host (41.6%), the 60.2 Kb region had a GC content of 47.7%. The significant difference in the GC% content, together with the fact that this region is flanked by short, direct repeats, led us to hypothesise that this DNA region had a heterologous origin and is present in the genome of *B. pumilus* 15.1, due to insertion or transposition events. Based on these results, we decided to study these two regions separately.

### 2.2. Genome Organisation of the 28.7 Kb Region

All the ORFs identified in the 28.7 Kb region were analysed with BLASTp to predict conserved protein domains and to determine the potential function of the putative proteins. The results of this analysis are detailed in Table 1. The predicted function for each ORF was determined with BLASTp unless otherwise specified. DNA and amino acid sequences of the identified ORFs are included in Appendix A. The 28.7 Kb region contained 40 predicted ORFs ranging from 141 bp (ORF 34) to 4272 bp (ORF 21). All of the ORFs, except for one (ORF 1), were encoded in the direct strand.

The 28.7 Kb region showed a similar genome organisation to PBSX, a well-characterised defective prophage from *Bacillus subtilis* 168 [33,34] that produce phage-like particles. Even the different cassettes present in PBSX defective phage (early, middle and late cassettes) could be easily identified in the 28.7 Kb region from *B. pumilus* 15.1 (PBSX-like region from now on) (Table 1). We predicted that the early cassette could be constituted by ORF 1, a backward-transcribed ORF, with a predicted Helix-Turn-Helix (HTH) conserved domain, similar to the Xenobiotic Response Element (XRE) transcriptional regulators (42.86% identity), the PBSX repressor responsible for maintaining the lysogenic state of the phage-like particle by repressing the transcription of the structural genes [38]. In addition, we predicted that the region from ORF 2 to ORF 6 could constitute the middle cassette, as putative proteins encoded by this region are related to DNA transcription. Although ORF 2, ORF 3 and ORF 4 showed no homology in the PBSX particle, ORF 6 showed a sigma-like factor conserved domain and 49% identity with Xpf, the RNA polymerase sigma factor of the PBSX particle. This protein is known to regulate the expression of the late cassette, and therefore, the expression of the structural genes of the PBSX particle [39]. ORF 2 showed an XRE family transcriptional regulator domain, and ORF 4 was homologous to proteins identified as Holliday junction resolvases, enzymes that both resolve the Holliday structures resulting from the recombination process, and are responsible for (i) debranching DNA structures in phages prior to the viral genome packaging into head particles and for (ii) degrading host DNA [40].

The late cassette, comprising all the structural genes that form the phage particle and genes responsible for lysis, was identified between ORF 7 and ORF 37. This region included the capsid protein (ORF 11), the tail sheath protein (ORF 17), the tail tube protein (ORF 18) and the baseplate J-like protein (ORF 26), among several others. In addition, a lysis module, with 3 ORFs (ORF 38-ORF 40) and with the same organisation as PBSX defective phage, was predicted at the 3´ end of the late cassette. ORF 38 and ORF 39 were identified as putative holins, proteins that accumulate and form aggregates to hydrolyse the host plasmatic membrane, and facilitate access to the endolysin (ORF 40) for the peptidoglycan degradation at the lysis stage [41,42].

### 2.3. The PBSX-like Region Is Highly Conserved in Bacteria Belonging to the B. pumilus Group

A search for homologous sequences to the PBSX-like region in NCBI database using Megablast, identified 38 complete genomes belonging to the *B. pumilus* group (Table 2) and two genomes from strains classified as *Bacillus* sp. These sequences were compared for Average Nucleotide Identity (ANI) with the JSpeciesWS algorithm, and their nucleotide synteny was determined with a dotplot (Figure 2).

The selected sequences were highly similar and showed extremely conserved synteny (Figure 2 and Figure 3), although, as expected, the level of synteny decreased in the more distant strains. Nucleotide similarity ranged between 87.08 to 96.15%, with an average similarity of 90.76%.

When the PBSX-like region from *B. pumilus* 15.1 was compared to *B. subtilis* defective phages PBSX and PBSZ, using Discontiguous Megablast algorithm, identity and cover sequence were 68.37% and 41% for PBSX and 68.72% and 41% for PBSZ. Using JSpeciesWS, the ANI observed was 66.13% for PBSX and 65.08% for PBSZ.

As shown in Figure 2, the synteny of all genes in PBSX-like regions, compared by dotplot, was highly conserved, except for two regions (marked with arrows); one at the middle (black arrow) and one at the 3´-end of the 28.7 Kb region (light grey arrow). Both variable regions were close to genes related to cell lysis. The first region (inside of the *B. pumilus* 15.1 ORF 21) was similar to the *xkdO* gene from PBSX, encoding a putative tape measure protein (TMP), a protein present in the tail tube, in fully functional and defective phages, that determines the tail length. TMP is known to be ejected after the irreversible binding of the phage to the host receptors and before the DNA injection in T4 phage (*E. coli* phage of Myoviridae family) [44]. TMP contains a conserved transglycosylase domain involved in peptidoglycan degradation. Some TMPs proteins have been proposed to participate in phage infection by hydrolysing the beta 1–4 glycosidic bond in the peptidoglycan, allowing the phage particle to inject the DNA into the host [45]. The variable region was found in the middle of ORF21, right outside of the transglycosylase domain, which is highly conserved and normally localised at the C-terminal end of the predicted protein. We could speculate that ORF 21 encodes a protein with transglycosylase activity at the C-terminal end and a variable domain to recognise specific molecules. In addition, the next ORF (ORF 22) encodes a protein with a LysM conserved domain and a peptidoglycan-binding domain, which is a homologue of XkdP (72% similarity) in PBSX.

The second variable region ranged from ORF 32 to ORF 37, a region that encodes putative proteins of unknown function but, given their position in the genome they could be related to proteins in the tail fibres [46]. Two of them, the ORF 31 (63 aa) and ORF 34 (46 aa), showed an Xkdx conserved domain of unknown function. These ORFs are normally found near holins and endolysins, so it is tempting to speculate that they could be part of the lysis module. In fact, ORF 38 encodes a holin, which surprisingly has more identity with Bhla holin from *B. subtilis* prophage SPBeta (100% cover and 77% similarity) than with the PBSX holin Xhla (40% cover and 34% similarity), an indication of the genomic plasticity that phages and phage-related particles have.

The ORF 32 to ORF 37 region did not show sequence homology in PBSX and constitute the most variable region in all the prophage genomes, with modifications and gene reorganisations that are not observed in other regions. This observation is in agreement with the idea of the region encoding tail fibres, as they are responsible for the recognition of the bacterial receptors that determine the specificity of the particle [47] (Figure 3).

### 2.4. Genome Organisation of the 60.2 kb Region

A comparative analysis of the 60.2 Kb region was also performed, to detect conserved protein domains and determine the potential function of the putative proteins. The result of this analysis is shown in Table 3. The predicted function for each ORF was determined by BLASTp, unless otherwise specified. The 60.2 Kb region contained 86 ORFs ranging from 141 bp (ORF 77) to 2805 bp (ORF 71). Most of the ORFs were encoded on the direct strand except for 10 ORFs (ORF 12, 13, 42, 47, 48, 56, 82, 83, 84 and 86). Six of these 10 reverse-transcribed genes are related to transcriptional regulation (ORF 12, 13, 82 and 84) or DNA integration/recombination (ORF 56 and ORF 86).

A close analysis of the 60.2 Kb region shows that the predicted ORFs can be grouped into two subregions according to their putative function. This separation does not imply that they have different origin, and it was only made for description purposes. The 5´end of the 60.2 Kb region (ORF 1 to 48; Table 3 shadowed in light grey; named subregion 1) of approximately 31.5 Kb, contained around 20 ORFs, with known functions related to DNA replication, DNA regulation and DNA metabolism; specifically, a DNA polymerase I (ORF 21), a helicase (ORF 10), a DNA primase (ORF 11), a Holiday junction resolvase (ORF 25), and several genes involved in nucleotide metabolism (ORF 31 to 33, ORF 36, ORF 38) and transcriptional regulation (ORF 2, ORF 12, ORF 13 and ORF 46).

The 3´ end subregion (ORF 49 to ORF 86; Table 3, not shadowed; named subregion 2), with a size of almost 28.7 Kb, contained phage-related proteins and ORFs encoding putative proteins with integrase/recombinase conserved domains (ORF 56 and ORF 86, both of which are backward-transcribed). Subregion 2 contains two ORFs encoding large and small subunits of the terminase (ORF 57 and ORF 58), a portal protein (ORF 59), capsid proteins (ORF 60 and 62), tail proteins (ORF 66, ORF 71 to 73), and a baseplate protein (ORF 75). In addition, putative proteins related to the host lysis can be observed, such as an N-acetylmuramoyl-L-alanine amidase (lysin) (ORF 79) and a holin (ORF 80). Finally, five ORFs related to DNA metabolism/regulation, four of which are transcribed backwards (ORF 82, ORF 83, ORF 84 and ORF 86), can be found at the 3′ terminal end of subregion 2. ORF 82 showed a conserved YoID domain with unknown function, but related to the UmuD subunit of Polimerase V in Gram-negative bacteria. ORF 84 and 85 (transcribed backward and forward, respectively) are two putative transcriptional regulators showing HTH conserved domains, and ORF 86 showed a conserved domain in proteins related to site-specific recombinases/integrases (XerD superfamily).

Both subregions 1 and 2 showed a similar GC content (47.84% for subregion 1 and 47.54% for subregion 2), so no differences in the phylogenetic origin are predicted. Determining the origin of the 60.2 Kb fragment could be a very useful piece of information, so it is planned to be addressed in the near future.

The 60.2 Kb region, found just downstream of the PBSX-like region, comprise 86 ORFs with no similar genome organisation to any other known phage, although PHASTER predicted this region to be part of a prophage, given that several phage-related putative proteins are encoded by this region, especially at the 3´end (subregion 2). The 60.2 Kb region was less common in bacterial genomes compared to the PBSX-like region, and only five strong coincidences were found in NCBI database, all present in genomes of strains belonging to the *B. pumilus* group. These strains were *B. pumilus* PDSLzg-1 (Accession number CP016784)*, B. cellulasensis* GLB197 (Accession number CP018574), *B. pumilus* NCTC10337 (Accession number LT906438), *B. altitudinis* P-10 (Accession number CP024204) and *B. aerophilus* strain 232 (Accession number CP026008.1). We compared the 60.2 Kb region with the five homolog sequences and found they were highly similar (82–89% identity) with an ANI of 84.17–87.6%. In addition, the 60.2 Kb region showed a conserved synteny in all the strains with no variable regions (Figure 4 and Figure 5).

The analysis of the 60.2 Kb region in the five *Bacillus* genomes showed that the region seems to be a DNA insertion, as it was predicted for *B. pumilus* 15.1, as (i) the CG content of the region is different from the host and (ii) this region is flanked by a direct repeat sequence that is part of a tRNA gene.

Comparative studies of all the strains containing the 60.2 Kb insert showed two different insertions sites: (i) The tRNA gene for Alanine, with a 45 bp direct repeat sequence (in *B. pumilus* 15.1 and *B. pumilus* NCTC10337) and (ii) the tRNA gene for Arginine, with a 48 bp direct repeat sequence in the other four strains (*B. altitudinis* P10, *B. aerophilus* 232, *B. cellulasensis* GLB197 and *B. pumilus* PDSLzg-1). The insertions, however, seem to reconstitute the tRNA sequence, so the function of these genes is presumably maintained in the six strains analysed.

The relative position between the 60.2 Kb region and the PBSX-like region in the six genomes was also studied (Figure 6). *B. pumilus* 15.1 and *B. pumilus* NCTC10337 showed the same organisation and insertion site, in the tRNA-Ala gene, next to PBSX-like region. Both of them shared the same short 45 bp direct repeat, which includes an important part of the tRNA sequence (data not shown); however, we observed differences downstream, where the region showed no homology between these two strains.

In *B. aerophilus* 232, *B. cellulasensis* GLB197, *B. altitudinis* P-10 and *B. pumilus* PDSLzg-1 the insertion was observed in a tRNA-Arg located ~300 Kb upstream the PBSX-like region. All of the strains showed the same insertion site and genomic organisation of the flanking areas.

### 2.5. Phylogenetic Analysis of the 98.1 Kb Region by Whole-Genome and Single-Gene Comparison

The phylogenetic relationships between bacteriophages are currently analysed by whole genome comparison [48] and single gene analysis. Different phage genes have been used for this last purpose; the major capsid protein [49], the integrase [50], or the large subunit of the terminase [51], among others. To determine the relationship between the prophage regions identified by PHASTER and previously known phages or phage-related structures, these two approaches were used.

For the whole-genome approach, all of the complete sequenced phages using *B. pumilus* as host and present in the Virus-host database (https://www.genome.jp/virushostdb/view/; last access 1 December 2020) were compared for nucleotide similarity by Average Nucleotide Identity and dotplot matrix comparison with the 28.7 and 60.2 Kb regions. No sequence similarity was found between them (data not shown). For that reason, a different phylogenetic study was conducted using the large subunit of the terminase, a protein that is commonly used for phage classification. Depending on the type of terminase, phages can be classified by their DNA packaging strategy, and the type of phage DNA ends [52] can be predicted. An analysis of the 98.1 Kb region identified by PHASTER rendered two sequences encoding a large subunit of terminases (ORF 8 in the PBSX-like region and ORF 57 in the 60.2 Kb region). Both sequences were compared to the 109 large terminase subunit sequences found in databases, some of which are from phages with well-known DNA packaging mechanisms (marked by * in Figure 7). As shown, most of the terminases from phages seemed to group together in four large clades, although not all of them were grouped (*Listeria* phage B054, *Paenibacillus* phage Emery, *Bacillus* phage SPBeta, *Bacillus* phage 0305 phi8-36, *Mycobacterium* phage Nigel, *Bacillus* phage phBC6A52, or *Campylobacter* phage Cpt1). Terminases from the seven well-defined DNA packaging strategies [52,53] grouped together in eight clades.

The headful packaging strategy was present in two clusters, the T4 like packaging system (Yellow), and the P22-like headful packaging system (Blue). Surprisingly, terminases from *Streptococcus* phage O1205 and sfi11 did not group in the main headful packaging P22-like clade. We observed differences in phylogenetic relationship between the two terminases found in *B. pumilus* 15.1 (marked with red square boxes). While the terminase found in the PBSX-like region (ORF8) clearly clustered into the P22-like HeadFul packaging group, where it formed a defined cluster with PBSX-like terminases, the terminase found in the 60.2 Kb region (tRNA-Ala insertion) (ORF 57) did not convincingly cluster in any group of well-characterised terminases and remained unclassified. These results suggested that both terminases are responsible for different DNA packaging strategies and support the hypothesis that they have an independent origin.

### 2.6. Isolation and Morphological Characterisation of Phage-Like Particles from B. pumilus 15.1

To demonstrate the functional capacity of the PBSX-like ORFs found in the genome of *B. pumilus* 15.1 to form phage-like particles, we performed an induction assay with mitomycin C. Seven hours after the addition of mitomycin C, an important decrease in the optical density of the *B. pumilus* 15.1 culture (final OD ≤ 0.2) was observed. Following induction, the lysate was examined by TEM, and phage-like particles with a typical Myoviridae morphology were observed (Figure 8A). The particles had very distinctive heads (39.53 ± 2.85 nm wide (n ≥ 3)), and a tail (227.34 nm ± 3.99 nm in length and 22.66 ± 1.81 nm in wide (n ≥ 3)) that seemed to be contractile, as two different forms of the particles, an extended form (white arrows) and a contracted form (black arrows), were observed. The contracted forms showed axial striations, whereas cross-striations were observed in the extended forms. The tails seemed to be comprised of an internal structure (11.53 ± 1.01 nm wide, black arrow head) and an external sheath (24.19 ± 2.76 nm wide, white arrow head) (Figure 8A,B). In addition, fibres were observed at the distal part of the tail (empty arrows). Some of the particles did not show the head, and only the tail was observed (empty arrow heads). The observed Phage-Like Particles obtained from a mitomycin C-induced *B. pumilus* 15.1 culture were called Bp15.1PLPs.

As Bp15.1PLP particles were extremely similar to the PBSX defective phage [32], it was tempting to speculate that those particles were encoded by the 28.7 Kb region found by PHASTER.

### 2.7. DNA Content in Bp15.1PLP Particles

After obtaining PEG-precipitated Bp15.1PLP particles from a mitomycin C-induced culture, the DNA present inside their capsids was isolated and visualised in an agarose gel. One single band around 9 Kb was observed, a size very different than the predicted for the PBSX-like region by *in silico* studies (28.695 Kb). The digestion of the 9 Kb DNA fragment with five different restriction enzymes (*Bam*HI, *Hin*dIII, *Sal*I, *Eco*RI and *Bgl*II) (Figure 9A) rendered no defined bands in most of the cases; instead, a DNA smear was obtained, which was especially patent in *Hin*dIII and *Eco*RI digestions, and seems to indicate the heterogeneity of the extracted DNA.

To identify the packed DNA in Bp15.1PLP, the *Eco*RI and *Bam*HI digested DNA was introduced in *Eco*RI and *Bam*HI digested pUC19 vectors, ligated and used to transform *E. coli* DH5α cells. Plasmid content from nine white colonies was sequenced using M13 F (5′-GTTTTCCCAGTCACGAC 3′) and M13 R (5′-CAGGAAACAGCTATGAC 3) primers from the vector.

Sequencing results showed that the cloned DNA belonged to different regions of the bacterial genome of *B. pumilus*15.1, randomly packaged into the Bp15.1PLP particle heads, without any relation to the prophage genome identified by PHASTER and with no homology with phage-like particle proteins. To rule out the possibility of bacterial DNA contamination in the purified particles, the experiment was repeated three times, with increasing concentration of DNAse after bacterial lysis and previous to Bp15.1PLP PEG-precipitation. The results were the same, so we concluded that the particle did not pack its encoding DNA, but random DNA from the host. In the bibliography, we further found that this phenomenon was also described in the defective phage PBSX [34].

### 2.8. Protein Characterisation of Bp15.1PLPs

To identify the major structural proteins present in Bp15.1PLP particle, a sucrose-gradient purified Bp15.1PLP suspension was subjected to 12% SDS-PAGE electrophoresis and stained with Coomassie brilliant blue. The Bp15.1PLP particle contained at least 15 proteins ranging from 110 to 17 KDa (Figure 9B). The most abundant bands were excised (shown with arrows) and identified by MALDI-TOF (Table 4). The results showed that all the Bp15.1PLP proteins identified by MALDI-TOF were encoded by the 98.1 Kb region under study. Proteins bands 1, 3 and 4 were encoded by ORFs present in the PBSX-like region, specifically ORF 17 (Phage tail sheath protein), ORF 11 (Phage major capsid protein) and ORF 18 (Phage tail tube protein). However, protein in band 2 was encoded by the ORF 62, identified as a major capsid protein by PHASTER (and as a Hypothetical Protein by Blastp) at the 60.2 Kb region (subregion 2), 41 Kb downstream the end of the PBSX-like region.

The surprising fact that Bp15.1PLP could be formed with proteins encoded from distant genes in the chromosome is similar to that which happens in other phage-like elements, the GTAs. The components of the most studied GTAs, RcGTA from *Rhodobacter capsulatus* and VSH-1 from *Brachyspira hyodysenteriae*, have a multilocus genome organisation and are encoded in five and two different loci, respectively, separated by hundreds of kilobases, and some of them with different evolutionary origins [54]. However, although this level of mosaicism is frequent in genomes, a simpler explanation could not be ruled out. Given that the PBSX-like region is highly distributed among the *Bacillus* genus does not make much sense that its production is dependent on the expression of a gene present in a DNA fragment scarcely found in bacterial genomes. Hence, the hypothesis of *B. pumilus* 15.1 having a second phage or phage-like particle, encoded by the 60.2 Kb region and also induced by mitomycin, was made. This hypothesis was supported by the fact that two different, non-phylogenetically related terminases were present in the 98.1 Kb fragment, one in the PBSX-like region and the other in the 60.2 Kb region.

### 2.9. B. pumilus 15.1 Produces a Phage That Is Co-Purified with Bp15.1PLP Particles

To test the hypothesis of a second phage-related particle produced by *B. pumilus* 15.1, TEM micrographs obtained from sucrose purified samples were closely inspected again, in search for less abundant phages or phages-like particles that could have been overlooked in the first analysis. The result of this close inspection was the identification of another phage structure, much less abundant than the Bp15.1PLP particle, which could be in accordance with our hypothesis (Figure 10). The relative abundance of this phage is also in agreement with the amount of protein found in SDS-PAGE analysis (band 2), identified as a major capsid protein by PHASTER, but being the least abundant of the selected proteins analysed by finger printing. The newly discovered particle showed a typical phage structure, with a 73.41 ± 0.15 (n = 3) nm hexagonal shaped head, highly electron-dense in the centre and clear at the edges, and a tail 155.03 ± 4.6 nm long and 12.01 ± 0.4 nm wide. The tail was slightly curved, non-contractile, showed cross-striation and seamed to end in a wider area (Figure 10C). Phage morphology was consistent with bacteriophages from the *Siphoviridae* family of the *Caudovirales* order (small heads and long flexible and non-contractile tails).

Given the fact that one of the proteins obtained in the sucrose purified lysates is produced by ORF 62 at the 60.2 kb region and that this region also contains two ORFs encoding large and small subunits of a terminase (ORF 57 and ORF 58), a portal protein (ORF 59), other capsid protein apart from the one codified by ORF 62 (ORF 60), tail proteins (ORF 66, ORF 71 to 73), a baseplate protein (ORF 75), putative proteins related to host lysis, such as an N-acetylmuramoyl-L-alanine amidase (lysin) (ORF 79) and a holin (ORF 80), make us draw the conclusion that this region could be responsible for the synthesis of a phage, as many of the required elements for phage assembly and function are present in this region.

After a bibliographic search, no publications reporting the production of phages in any of the five strains containing the 60.2 Kb homolog region in *B. pumilus* 15.1 (*B. pumilus* PDSLzg-1, *B. cellulasensis* GLB197, *B. pumilus* NCTC10337, *B. altitudinis* P-10 and *B. aerophilus* strain 232) were found (May 2021). To search for similar phages that the one observed in *B. pumilus* 15.1 lysates, an extensive search was conducted using the Viral Genome Database at NCBI (https://www.ncbi.nlm.nih.gov/genome/viruses/; last access 1 May 2021). This database compiles all the known and currently available viral genomes. In this database, there are 143 completely sequenced phage genomes (plus 2 incomplete) that have been either found in a *Bacillus* strain or which can use a *Bacillus* strain as a host. The *Bacillus* species with the highest number of genome-sequenced phages were *B. thuringiensis* (44 phages), *B. cereus* (31 phages), *B. subtilis* (21 phages), *B. megaterium* (17 phages), *B. pumilus* (14 phages) and *B. anthracis* (3 phages) (Table 5). Other *Bacillus* species were hosts for only one phage, and 7 phages used unclassified *Bacillus* species (*Bacillus* sp.) as a host. The size of *Bacillus* phages ranged from 18,379 nt (phage BsuP-Goel) from *B. subtilis* to 497,513 nt (phage G) from *B. megaterium* (data not shown). Phages found in *B. pumilus* (Table 6) showed a genome size ranging from 18,466 nt (phage Bpu_PumA1) to 160,281 nt (phage Bobb from *B. pumilus* SAFR32). The size of the 60.2 Kb inserted fragment found in *B. pumilus* 15.1 lies into this size range.

A whole sequence comparison of the 60.2 Kb region found in *B. pumilus* 15.1 with the 14 phage genome sequences found in *B. pumilus* was performed using tBlastx and plotted by Easyfig. No relevant similarity (nor genomic synteny) of the 60.2 Kb region with any of the previously described *B. pumilus* phages was found after the analysis. This result supports the hypothesis that the 60.2 Kb region encodes for a novel *B. pumilus* phage, named Bp15.1Hope. The characterisation of the novel phage and the search for a suitable host has been already undertaken in our lab.

### 2.10. Bacteriocin-Like Activity of PEG-Precipitated Phage like Particles

It has been reported that the *B. subtilis* PBSX particle has a bacteriocin-like activity against host-related strains, which seems to play an important role for the strain in competing with other bacteria in the same environment [32]. To determine whether the Bp15.1PLP particle has similar bacteriocin-like activity, fifteen strains from our laboratory collection were tested as described in the Materials and Methods section. Several strains were susceptible to Bp15.1PLP particles (Table 7), showing a clear halo when a drop of a suspension of Bp15.1PLP (and Bp15.1Hope) was placed over the strain in a spot test. All of the susceptible strains were *B. pumilus* strains, including *B. pumilus* 15.1C, an extra-chromosomal DNA cured strain obtained from *B. pumilus* 15.1 and previously described by our group [5]. Bp15.1Hope was not able to form plaques in any of the strains tested in lawn assays (data not shown), so the halos observed seem to indicate that Bp15.1PLP has a strong bacteriocin-like activity. This activity must be considered very narrow, as only strains from the same species were susceptible. This specificity agrees with the previously described diversity in the tail fibre sequences found in Bp15.1PLP, as fibres are responsible for recognising bacterial receptors.

Moreover, it was observed that Bp15.1PLP was not active against *B. pumilus* 15.1 strain, the strain that produces the particles. This is a common feature among phages, as lysogenic hosts are not susceptible to the lytic action of the phage that they produce, and apparently, Bp15.1PLP has the same behaviour. Surprisingly, the closely related *B. pumilus* 15.1C strain, previously obtained by treatment with acridine orange for plasmid curing, is susceptible to the action of Bp15.1PLP particles. This phenomenon will be further investigated in the near future as it is not straight-forward to understand how curing plasmids in a strain can make them susceptible to the action of their own producing PLP.

### 2.11. Bioassaying Ceratitis Capitata Larvae with Phage like Particles

Phage-related systems, such as the *Photorabdus* virulence cassette (PVC) and the anti-feeding prophage (AFP) from *Serratia*, have been previously described to be the delivery system for virulence factors in entomopathogenic bacteria [25,26]. To study the contribution of Bp15.1PLP and Bp15.1Hope in the toxicity of *B. pumilus* 15.1 toward *C. capitata*, diet contamination bioassays using first-instar larvae were performed. A Bp15.1PLP/Bp15.1Hope preparation consisting of a mitomycin-induced lysate culture was bioassayed in combination or not with a *B. pumilus* 15.1 sporulated culture (bioassayed in sub-lethal conditions). The rationale behind this experiment was that if phage particles are somehow involved in *C. capitata* toxicity, the increase in the number of particles and the induction of the genes related to the particles would have a synergistic effect on the toxicity of a *B. pumilus* 15.1 sporulated culture. Particle preparation showed no toxicity against *C. capitata* (even when it was 200 times concentrated) (Table 8), and no synergistic effect between the sporulated culture and the phage particles was observed. These results seem to suggest that neither Bp15.1PLP nor Bp15.1Hope are related to *B. pumilus* 15.1 toxicity against *C. capitata.*

### 2.12. Concluding Remarks

The 98.1 kb DNA region, found in Contig 59 of the entomopathogenic strain *B. pumilus* 15.1, encodes a PBSX-like particle, called Bp15.1PLP, and an undescribed *B. pumilus* phage named Bp15.1Hope. The Bp15.1PLP particle is encoded by a 28.7 kb region, and Bp15.1Hope seems to be a genomic insertion in a tRNA gene. The Bp15.1PLP particle packages bacterial host DNA in its head, and shows bacteriocin activity toward all of the *B. pumilus* strains tested, but not to other related bacteria (such as *B. subtilis*). The Bp15.1Hope phage is not similar to any of the currently described *B. pumilus* phages, and it can be classified as a *siphovirus* according to its morphology. Both particles are expressed in liquid cultures when the strain *B. pumilus* 15.1 is induced by mitomycin, but at a very different rate, as Bp15.1PLP is much more abundant than Bp15.1Hope. None of these particles is related to the entomopathogenic activity that *B. pumilus* 15.1 shows toward *C. capitata.* The separation of these two particles and DNA sequencing of Bp15.1Hope have become our next objectives to definitely prove that Bp15.1Hope particles are produced by the 60.2 kb region.

## 3. Materials and Methods

### 3.1. Bacterial Strains and Growth Conditions

The following bacterial strains were used in this study: *Bacillus pumilus* 15.1 [1], *B. pumilus*15.1_C [2] (extra-chromosomal-cured *B. pumilus* 15.1 strain), *B. pumilus* M1, *B. pumilus* M2, (both kindly provided by Dr C. Calvo) [55], *B. pumilus* 8A3 (ATCC 7061), (kindly provided by Dr C. Berry, Cardiff University), *B. thuringiensis* 78/11, *Bacillus subtilis* 168C, *B. cereus* 169, *Micrococcus* sp., *Arthrobacter* sp., *Lactococcus garvieae*, *Lactococcus* NZ9000 and *Listeria innocua* (CECT 4030) (kindly provided by Dr Martinez Bueno, Granada University).

Luria-Bertani (LB) medium broth or agar was routinely used for growing bacteria at 30 °C. Brain-Heart Infusion (BHI) broth was used for growing Lactococcus, and Listeria strains at 30 °C. When sporulation was required, T3 medium [56] was used, and the culture was incubated at 30 °C for 72h.

### 3.2. Induction, Concentration and Purification of Phage-Like Particles

Phage-like particles were directly obtained from *B. pumilus* 15.1 strain by adding mitomycin C to the bacterial culture. Fresh LB broth was inoculated with an overnight culture (1:100 dilution) and incubated at 30 °C until the culture reached an optical density at 600 nm (OD_600_) of 0.4. Then, mitomycin C (0.5 µg/mL final concentration) was added, and the culture was incubated until lysis occurred (OD_600_ <0.2). Two percent (*v/v*) chloroform was added to the lysate and incubated for 20 min under agitation.

For the rapid concentration of phage-like particles, when no extra purification was necessary, the lysate was centrifuged at 4000× *g* for 15 min to remove cell debris. The supernatant was then filtered through a 0.2 µm nitrocellulose filter and treated with 10 µg/mL DNAse and RNAse for 30 min. Finally, the supernatant was centrifuged at 100,000× *g* for 90 min to pellet phage-like particles. The pellet was resuspended in 1/200 of the initial volume of the culture in SM medium (NaCl 0.1 M; MgSO_4_ 0.1 M; Tris 0.05 M; Gelatine 0.01%; pH 7.5).

Phage-like particles were also concentrated and partially purified by a polyethylene glycol (PEG) precipitation method according to [57] with minor modifications. Briefly, lysates were centrifuged twice at 4000× *g* for 30 min to remove cell debris. Lysates were treated with 1 µg/mL DNAse and RNAse at 37 °C for 30 min, and then, solid NaCl was added to a final concentration of 1 M. Lysates were incubated for 1 h on ice and centrifuged at 12,000× *g* for 30 min. To pellet phage-like particles, 10% *w/v* PEG 8000 was added to the supernatant, incubated on ice for at least 1 h, and then centrifuged at 12,000× *g* for 30 min. Pellets were resuspended in 1/400 of the initial culture volume in SM medium, vortexed with the same volume of chloroform and centrifuged for 10 min at 16,000× *g*. The upper aqueous phases, containing the phage-like particles, were collected and stored at 4 °C until use.

When necessary, particle suspension was placed on a preformed discontinuous sucrose gradient (60-67-75 and 83%) and ultra-centrifuged (53,000× *g*, 4 °C) in a Beckman JSW24.38 rotor for 16 h. Fractions were collected from the top of the gradient by pipetting, washed three times in SM buffer and analysed by SDS-PAGE and Transmission Electronic Microscopy (TEM) to observe proteins and phage-like particles, respectively.

### 3.3. Transmission Electron Microscopy (TEM)

A *B. pumilus* 15.1 culture was induced with mitomycin C and centrifuged (4000× *g* 15 min 4 °C) when lysis was observed. The supernatant was filtered through a 0.2 µm nitrocellulose filter and sent to the Microscopy service to be fixed and negatively stained with 1% uranyl acetate. Samples were examined under a ZEISS EM 902 transmission electron microscope (CIC, Universidad de Granada, Granada, Spain).

High-titre phage-like samples obtained by sucrose gradient, ultracentrifugation (100,000× *g* for 1 h) or PEG 8000 precipitation were diluted and also analysed by TEM to check the structural integrity of phage-like particles.

### 3.4. Bacteriocin-like Activity

The bacteriocin-like activity of phage-like particles was carried out in a spot test, with some modifications [58]. Overnight cultures (12–16 h) of the test bacteria were streaked on an LB plate with the help of a sterile toothpick; once dried, a drop of phage-like particle suspension, after PEG-precipitation, was placed on one end of the streak, incubated O/N at 37 °C and observed for clear zones on the following day.

### 3.5. Bp15.1PLP DNA Isolation

DNA extraction from sucrose-purified phage-like particles was performed using the DNeasy Blood and Tissue DNA kit (Qiagen) according to the manufacturer’s instructions from a maximum of 200 µL of a phage-like particle suspension.

### 3.6. Phage-Like Particle DNA Mapping and Sequencing

The DNA contained in the phage-like particles was isolated as described above and digested with restriction enzymes (NEB) according to the supplier’s instructions. Digestions were analysed in a 0.8% *w/v* agarose gel. *Hin*dIII and *Eco*RI fragments were ligated to *Hin*dIII or *Eco*RI-digested pUC19 vectors and introduced in *E. coli* DH5α competent cells by heat shock [59].

The resulting transformants were screened by toothpick assay [60]. Positive clones were isolated, plasmids extracted and sequenced with M13 Forward and Reverse primers at the Genomic Service of Instituto de Parasitología y Biología Molecular Lopez Neyra, Granada, Spain.

### 3.7. Phage-Like Particle Analysis by SDS-PAGE and Protein Identification

Ten microlitres of a suspension of sucrose-purified phage-like particles were analysed in a 12% wt/v SDS-PAGE gel and stained with Coomassie brilliant blue according to standard procedures. PAGE Ruler Plus Pre-Stained Protein Ladder (Thermo Scientific) was used as size protein markers.

Protein bands were excised and digested with trypsin before identification by MALDI-TOF. The data obtained were analysed by MASCOT at the Centro de Biología Molecular Severo Ochoa, Madrid, Spain.

### 3.8. Ceratitis Capitata Larval Bioassay

*C. capitata* bioassays were performed as described previously by our group [1] with some modifications. A *B. pumilus* 15.1 sporulated culture was 40-fold concentrated by lyophilisation, and the lysate obtained after mitomycin C induction was used directly or after PEG concentration. To determine whether the phage-like particles had some synergistic effect on the toxicity of the bacteria, both phage-like particles and bacteria were bioassayed together. Bacterial or phage-like samples were mixed with the artificial *C. capitata* larva diet in a diet contamination bioassay. Deionised water and SM buffer were used as a negative control in the bioassays. All bioassays were performed at least twice from separate cultures under the conditions described previously by our group [1].Mortality was recorded 14 days after the beginning of the bioassay. Results were statistically analysed using a one-way ANOVA method.

### 3.9. Bioinformatics

*B. pumilus* 15.1 draft genome sequence (Accession number LBDK00000000.1) was analysed with PHASTER algorithm [35] (PHAge Search Tool Enhanced Release http://phaster.ca, University of Alberta, last access date 1 September 2019) for potential prophage detection. Predicted phage sequences were analysed by comparative analysis with BLASTp [61] against the total NCBI protein database. tRNA identification was performed using tRNAscan-SE [62]. DotPlot comparison was performed with Gepard 1.41 [43], using a word size of 10, and the average nucleotide identity based on Blast+ (ANIb) analysis was performed with JspeciesWS software [63]. The neighbour-joining tree was constructed using ClustalW [64] and MEGAX [65], following the Poisson model with a bootstrapping of 1000 trials.

## Figures and Tables

**Figure 1 ijms-22-08164-f001:**
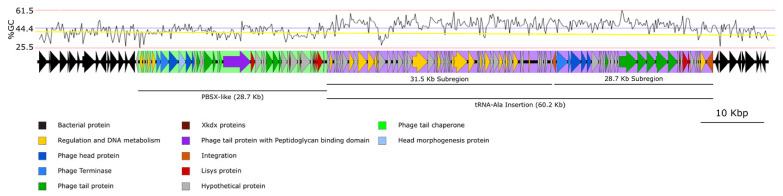
Schematic representation of the genomic area detected by PHASTER (coloured) and flanking regions (black). The PBSX-like particle is highlighted in green, while the tRNA-Ala is marked in purple. The subregions found in the tRNA-Ala insertion are also indicated. The GC content is showed at the top of the figure. The blue line indicates the mean GC content of the 98 Kb fragment, and the yellow line shows the average GC content for *B. pumilus*. The potential function of each ORF was inferred by Blastp.

**Figure 2 ijms-22-08164-f002:**
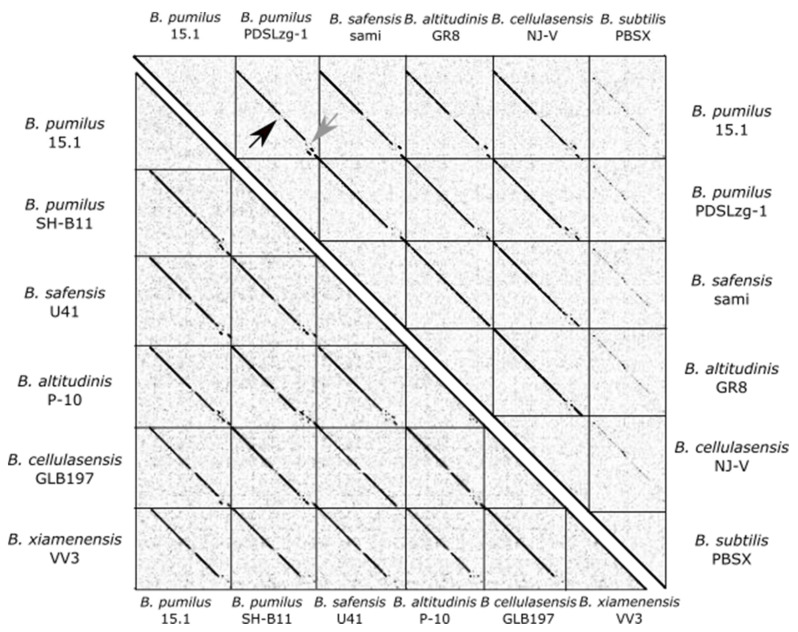
Whole-genome dot-plot comparison of some PBSX-like particles sequences from bacteria belonging to the *B. pumilus* group (shown in bold in Table 2). A comparison was made with Gepard 1.40 [43] with a word length of 10 nucleotides.

**Figure 3 ijms-22-08164-f003:**
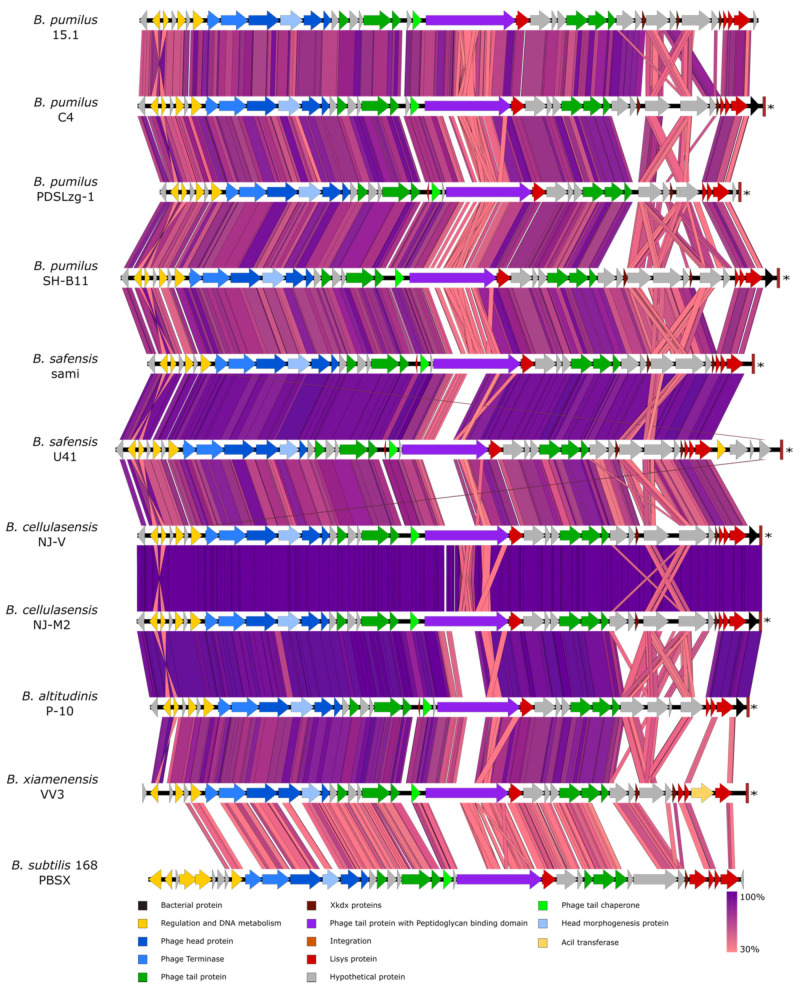
Comparative analysis of the PBSX-like regions among different *B. pumilus* related species. The potential function of each ORF was inferred by Blastp and shown as coloured arrows. tRNAs are indicated by asterisks. Homology studies were made by tBlastx and plotted with Easyfig with a minimal length of 50 bp.

**Figure 4 ijms-22-08164-f004:**
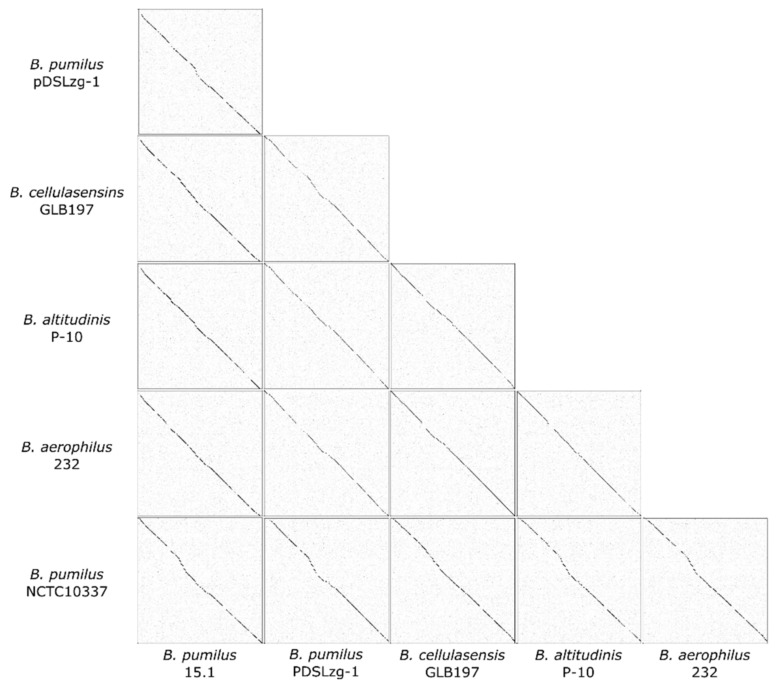
Comparative analysis of the 60.2 Kb region present in *B. pumilus* related species by dotplot. In *B. pumilus* 15.1 and *B. pumilus* NCTC10337 the 60.2 Kb was found inserted in a tRNA-Ala gene, while in *B. cellulasensis* GLB197, *B. altitudinis* P-10, *B. aerophilus* 232 and *B. pumilus* PDSLzg-1 the insertion was observed in a tRNA-Arg gene.

**Figure 5 ijms-22-08164-f005:**
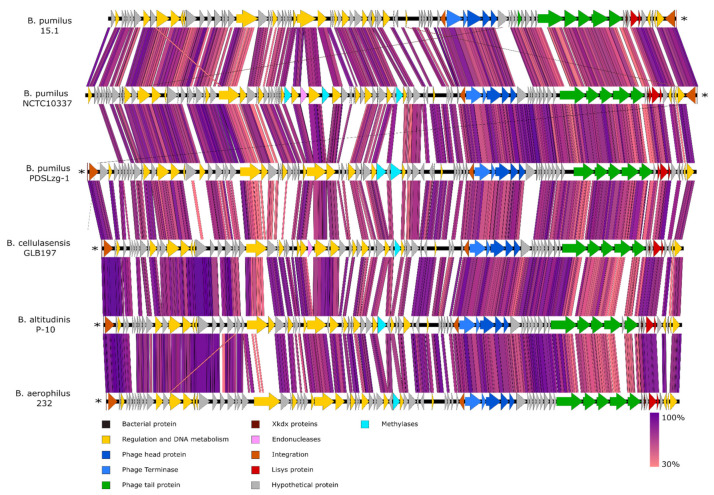
Comparative analysis of the 60.2 Kb insertion fragment among different *B. pumilus* related species. The potential function of each ORF was inferred by Blastp and shown as coloured arrows. tRNAs position are indicated by asterisks. Homology studies were made by tBlastx and plotted with Easyfig with a minimal length of 50 bp.

**Figure 6 ijms-22-08164-f006:**
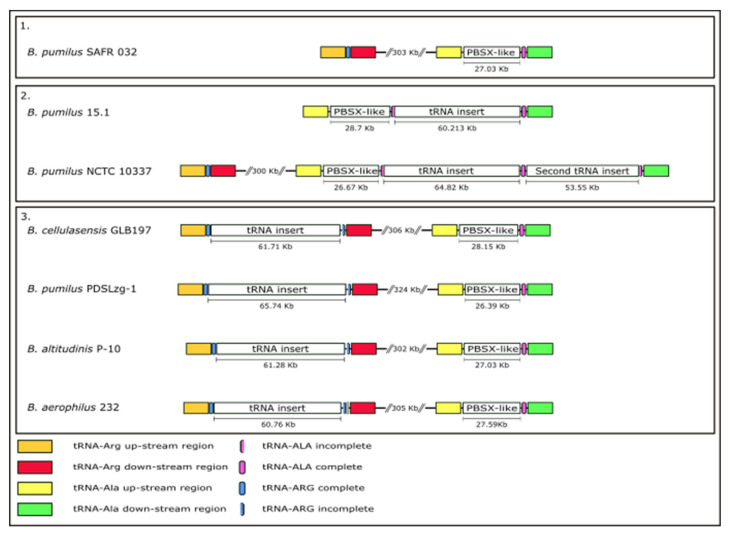
Genomic organisation comparison of the 60.2 Kb insertion in *B. pumilus* related strains. The relative position of the tRNA inserts and PBSX-like particles are shown in the six strains and compared to the model strain *B. pumilus* SAFR 032 (panel 1), which lacks tRNA inserts. tRNA-Arg gene is shown in blue, and tRNA-Ala is shown in pink. In most *Bacillus* species (panel 1), the PBSX-like region is adjacent to the tRNA-Ala. In *B. pumilus* 15.1 and NCTC 10337 (panel 2), tRNA insert is adjacent to PBSX-like region, in the tRNA-Ala gene. However, in *B. cellulasensis* GLB197, *B. pumilus* PDSLzg-1, *B. altitudinis* P-10 and *B. aerophilus* 232, inserts are present in the tRNA-Arg gene, *circa* 300 bp upstream PBSX-like genome (panel 3).

**Figure 7 ijms-22-08164-f007:**
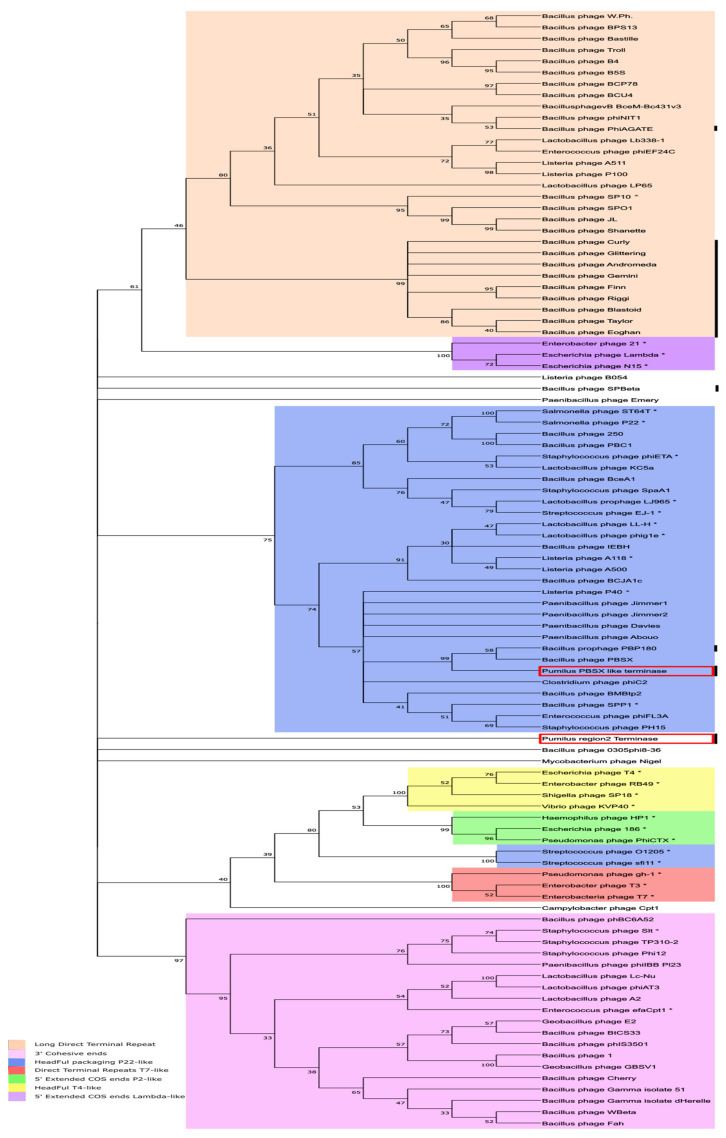
Neighbor-joining tree of the large terminase subunit in different phage and phage related structures. The amino acid sequences of large subunit terminases from 99 phages were aligned with ClustalW, and a neighbor-joining tree was prepared with MEGA X, with bootstrapping set to 1000. The phylogenetic relationship was inferred by the Poisson model. The numbers near bifurcations are bootstrap percentages for 1000 trials, and all the branches with a bootstrapping level of less than 30% were collapsed. Phages whose DNA termini was reported are marked with *, and clusters of similar packaging methods are highlighted in different colours marking previously clustered terminases in [51,52], unclustered terminases are not highlighted. *B. pumilus* phages are marked by black-vertical lines, and the terminases from *B. pumilus* 15.1 sequence are highlighted by red boxes. Phages are named according to the EBI virus database and virus hostdb.

**Figure 8 ijms-22-08164-f008:**
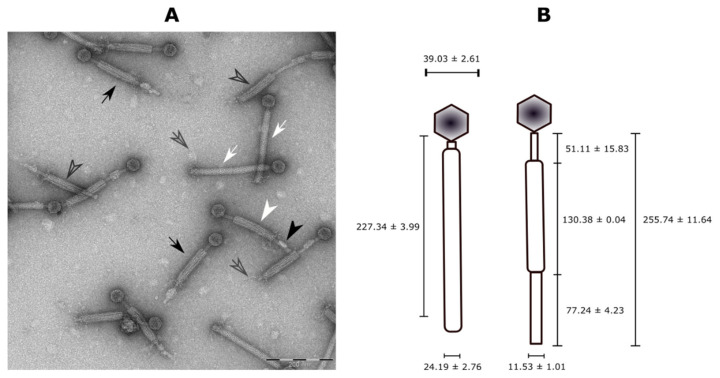
(**A**) Transmission Electron Microscopy (TEM) of partially purified Bp15.1PLPs by sucrose gradient. Two different forms were observed, an extended particle (white arrows) and a contracted form (black arrows). In the contracted form, a tail tube (black arrowhead), a tail sheath (white arrowhead) and fibres at the end of the tails (empty arrows) were distinctly observed. Some of the particles lacked a head (empty arrowhead). (**B**) Schematic representation of the observed phage-like particles with the average size of each part (obtained from at least three measurements).

**Figure 9 ijms-22-08164-f009:**
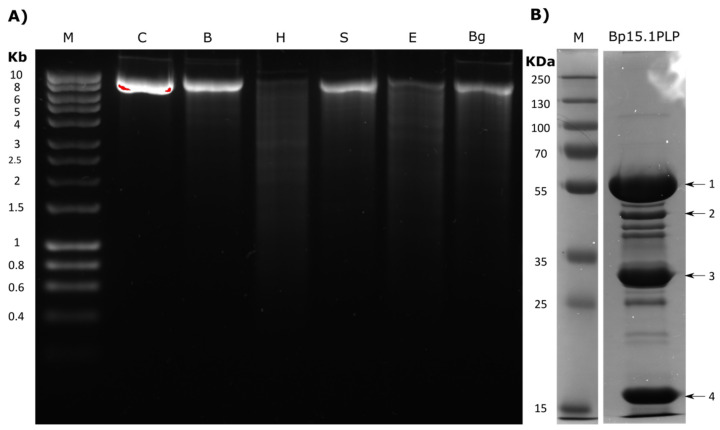
(**A**) Agarose gel electrophoresis of the DNA contained in Bp15.1PLP before and after restriction enzyme digestion. The nucleic acids were extracted from partially purified Bp15.1PLPs after mitomycin C induction and DNAse treatment. The DNA was digested with *Bam*HI B), *Hin*DIII (H), *Sal*I (S), *Eco*RI (E) and *Bgl*II (Bg) and compared to the control without enzymatic digestion (C). HyperLadder™ 1 kb (BioLine) (M) was used as a weight marker. (**B**) SDS-PAGE analysis of partially purified Bp15.1PLPs particles along with the PageRuler Plus Prestained Protein Ladder (ThermoFisher) (M). The most abundant proteins (arrows 1 to 4) were excised and identified by MALDI-TOF.

**Figure 10 ijms-22-08164-f010:**
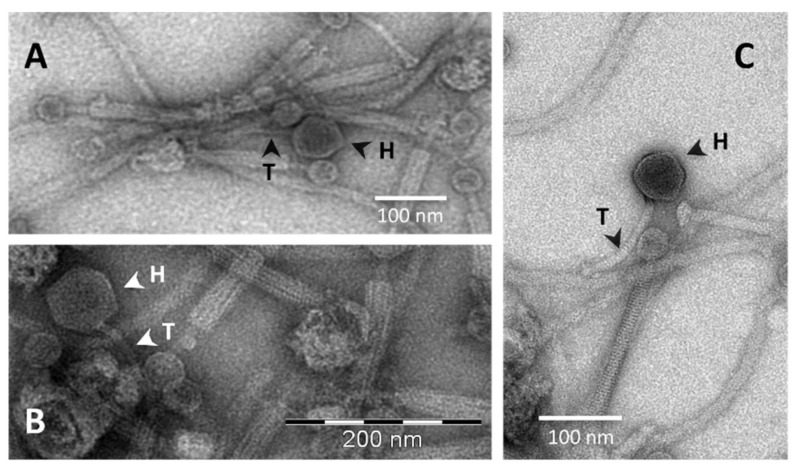
Transmission Electron Microscopy of partially purified (sucrose gradient) bacteriophage-like particles. Apart from Bp15.1PLP, another less abundant phage-like particle was observed. Heads (H) and tails (T) of this particle are shown with arrowheads.

**Table 1 ijms-22-08164-t001:** Predicted ORFs in the 28.7 Kb region identified by PHASTER.

ORF	Direction	Predicted Nº of aa	Predicted Function	Module
1	-	125	XRE family transcriptional regulator	Early
2	+	87	XRE family transcriptional regulator	
3	+	75	Hypothetical protein	
4	+	133	Holliday junction resolvase	Middle
5	+	68	Hypothetical protein XtrA Domain	
6	+	172	Positive control sigma-like factor	
7	+	214	Small subunit of the terminase*	
8	+	437	Phage terminase large subunit	
9	+	485	portal protein superfamily	
10	+	362	Putative phage serine protease	Late
**11**	**+**	**307**	**Phage major capsid protein**	
12	+	127	Head-tail connector protein	
13	+	118	Tail shaft protein	
14	+	165	Putative tail-component	
15	+	151	Hypothetical protein	
16	+	70	Hypothetical protein	
**17**	**+**	**448**	**Phage tail sheath protein**	
**18**	**+**	**147**	**Phage tail tube protein**	
19	+	73	Hypothetical protein	
20	+	146	Tail assembly chaperone protein	
21	+	1423	TMP with transglycosylase domain	
22	+	222	LysM peptidoglycan binding domain	
23	+	336	Hypothetical protein	
24	+	88	DUF2577 Protein of unknown function	
25	+	140	DUF 2634Protein of unknown function	
26	+	349	baseplate J like protein	
27	+	307	DUF 2313 Family of phage tail proteins	
28	+	127	DUF3751 Phage tail-collar fibre protein	
29	+	290	Hypothetical protein	
30	+	109	XkdW protein	
31	+	63	XkdX protein	
32	+	371	PHA01818 superfamily	
33	+	106	Hypothetical protein	
34	+	46	XkdX protein	
35	+	470	Hypothetical protein	
36	+	115	Putative short tail fibre *	
37	+	48	XkdX protein	
38	+	70	Holin BhlA	
39	+	87	Holin	
40	+	267	N-acetylmuramoyl-L-alanineamidase	

* Function predicted by PHASTER. +/− Indicate Forward/Backward direction of transcription, respectively. Proteins found in fingerprinting analysis are shown in bold.

**Table 2 ijms-22-08164-t002:** Strains and accession number of the highest hits were obtained to compare the 28.7 kb region from *B. pumilus* 15.1 with sequences at the Genbank. The coverage and identity of the found regions are also indicated. Genomes selected for whole-genome dot-plot comparison are shown in bold.

	Organism	Cover	Identity	Accession Number
1	*Bacillus* sp. WP8	0.86	98.01%	CP010075.1
2	*Bacillus safensis* strain U17-1	0.85	97.14%	CP015611.1
**3**	***Bacillus safensis* strain U41**	**0.85**	**97.14%**	**CP015610.1**
4	*Bacillus safensis* strain KCTC 12796BP	0.85	95.98%	CP018197.1
5	*Bacillus safensis* strain U14-5	0.79	95.95%	CP015607.1
6	*Bacillus safensis* strain FO-36b	0.84	95.93%	CP010405.1
7	*Bacillus safensis* strain BRM1	0.83	95.58%	CP018100.1
**8**	***Bacillus safensis* strain sami**	**0.85**	**95.52%**	**CP032830.1**
9	*Bacillus safensis* strain pgKB20	0.89	95.52%	CP043404.1
10	*Bacillus safensis* strain IDN1	0.87	94.79%	AP021906.1
11	*Bacillus altitudinis* strain Cr-1	0.88	87.44%	CP031774.1
12	*Bacillus pumilus* strain C4	0.87	87.30%	CP011109.1
**13**	***Bacillus pumilus* strain SH-B11**	**0.94**	**86.86%**	**CP010997.1**
**14**	***Bacillus pumilus* strain PDSLzg-1**	**0.81**	**92.32%**	**CP016784.1**
15	*Bacillus pumilus* strain ZB201701	0.83	92.30%	CP029464.1
16	*Bacillus pumilus* strain MTCC B6033	0.89	87.92%	CP007436.1
**17**	***Bacillus altitudinis* strain P-10**	**0.79**	**87.91%**	**CP024204.1**
18	*Bacillus altitudinis* strain GQYP101	0.85	87.84%	CP040514.1
19	*Bacillus pumilus* strain NCTC10337	0.83	91.72%	LT906438.1
20	*Bacillus altitudinis* strain W3	0.86	87.80%	CP011150.1
21	*Bacillus* phage PBP180	0.81	88.05%	KC847113.1
22	*Bacillus pumilus* strain 145	0.82	91.54%	CP027116.1
23	*Bacillus cellulasensis* strain ku-bf1	0.77	87.68%	CP014165.1
**24**	***Bacillus cellulasensis* strain NJ-V**	**0.82**	**87.63%**	**CP012330.1**
25	*Bacillus cellulasensis* strain NJ-V2	0.82	87.63%	CP012482.1
26	*Bacillus cellulasensis* strain NJ-M2	0.82	87.63%	CP012329.1
**27**	***Bacillus cellulasensis* strain GLB197**	**0.83**	**87.61%**	**CP018574.1**
28	*Bacillus pumilu*s strain TUAT1	0.83	87.61%	AP014928.1
29	*Bacillus aerophilus* strain 232	0.84	87.59%	CP026008.1
30	*Bacillus altitudinis* strain SGAir0031	0.86	87.58%	CP022319.2
**31**	***Bacillus altitudinis* strain GR-8**	**0.83**	**87.67%**	**CP009108.1**
32	*Bacillus altitudinis* strain HQ-51-Ba	0.77	87.56%	CP040747.1
33	*Bacillus pumilus* strain SH-B9	0.78	91.26%	CP011007.1
34	*Bacillus altitudinis* strain CHB19	0.77	87.42%	CP043559.1
35	*Bacillus pumilus* strain SF-4	0.78	91.01%	CP047089.1
36	*Bacillus pumilus* strain SAFR-032	0.8	90.94%	CP000813.4
37	*Bacillus pumilus* strain 150a	0.75	90.93%	CP027034.1
**38**	***Bacillus xiamenensis* strain VV3**	**0.77**	**88.58%**	**CP017786.1**
39	*Bacillus altitudinis* strain FD48	0.8	87.77%	CP025643.1
40	*Bacillus* sp. ms-22	0.82	88.48%	CP046653.1

**Table 3 ijms-22-08164-t003:** Predicted ORFs in the 60.213 Kb region identified by PHASTER.

ORF	Direction	Predicted Nº of aa	Predicted Function
1	+	150	Hypothetical protein
2	+	138	Transcriptional regulator Helix-turn-helix domain
3	+	121	Hypothetical protein
4	+	75	Hypothetical protein
5	+	125	Hypothetical protein
6	+	122	Hypothetical protein
7	+	269	Hypothetical protein
8	+	253	DNA replication protein with ATP binding domain
9	+	216	Hypothetical protein
10	+	463	Phage DNA helicase
11	+	333	DNA primase
12	−	104	XRE family transcriptional regulator
13	−	68	XRE family transcriptional regulator
14	+	388	Hypothetical protein
15	+	176	Hypothetical protein
16	+	167	Hypothetical protein
17	+	257	Hypothetical protein
18	+	83	Hypothetical protein
19	+	144	Hypothetical protein
20	+	237	Hypothetical protein
21	+	759	DNA Polimerase I
22	+	349	Hypothetical protein
23	+	84	Hypothetical protein
24	+	65	Hypothetical protein
25	+	184	Holiday junction resolvase
26	+	164	Hypothetical protein
27	+	87	Hypothetical protein
28	+	121	Hypothetical protein
29	+	67	Hypothetical protein
30	+	64	Hypothetical protein
31	+	118	Class 1b ribonucleoside-diphosphate reductase assembly flavoprotein NrdI
32	+	695	Class 1b ribonucleoside-diphosphate reductase subunit alpha
33	+	324	Class 1b ribonucleoside-diphosphate reductase subunit beta
34	+	64	Hypothetical protein
35	+	70	Hypothetical protein
36	+	108	Deoxyuridine 5’-triphosphate nucleotide hydrolase
37	+	166	Hypothetical protein
38	+	264	FAD-dependent thymidylate synthase
39	+	164	This protein contain the critical active site aspartate of mltA-like lytic transglycosylases
40	+	283	Hypothetical protein
41	+	186	Dephospho-CoA kinase
42	−	124	Hypothetical protein
43	+	124	DUF134 domain
44	+	68	Hypothetical protein
45	+	162	Hypothetical protein
46	+	278	MarR family transcriptional regulator
47	−	61	Hypothetical protein
48	−	127	Hypothetical protein
49	+	70	XRE family transcriptional regulator
50	+	93	Hypothetical protein
51	+	93	Hypothetical protein
52	+	95	Hypothetical protein
53	+	73	Hypothetical protein
54	+	94	Hypothetical protein
55	+	100	Hypothetical protein
56	−	181	Phage integrase family
57	+	588	Large subunit terminase
58	+	141	Hypothetical protein with HTH TNP domain
59	+	533	Phage portal protein
60	+	276	Minor capsid protein *
61	+	227	Scaffold protein *
**62**	**+**	**363**	**Major capsid protein ***
63	+	73	Hypothetical protein
64	+	128	Hypothetical protein
65	+	112	Hypothetical protein
66	+	136	Bacteriophage putative tail-component
67	+	124	Hypothetical protein
68	+	180	Hypothetical protein
69	+	126	Hypothetical protein
70	+	71	Hypothetical protein
71	+	934	Phage tail tape measure protein
72	+	475	Phage tail protein
73	+	463	Prophage endopeptidase tail
74	+	626	right-handed parallel beta-helix repeat-containing protein
75	+	483	Phage baseplate upper protein
76	+	95	Hypothetical protein
77	+	46	XkdX family protein
78	+	103	Hypothetical protein
79	+	277	N-acetylmuramoyl-L-alanineamidase
80	+	79	Phage holin
81	+	183	Hypothetical protein
82	−	114	YOID-like family protein
83	−	76	Hypothetical protein
84	−	71	HTH transcriptional regulator
85	+	263	phage protein HTH domain containing
86	−	353	Site specificrecombinase/integraseXerD

* Function predicted by PHASTER. +/− Indicate Forward/Backward direction of transcription, respectively. The 31.5 Kb subregion (shadowed) and 28.7 Kb subregion (not shadowed) are indicated in the table. The protein found in fingerprinting analysis is shown in bold.

**Table 4 ijms-22-08164-t004:** Proteins identified by MALDI-TOF.

						Protein Function	
Band	Mass (kDa)	MASCOT Identification	Phaster ORF	Score	Nº Matches	By Phaster	By Blastp
1	49,06	Phage portal protein [*Bacillus pumilus*]	PBSX-like ORF 17	86	9	XkdK-like tail sheath protein	XkdK-like tail sheath protein
2	40,12	Hypothetical protein [*Bacillus* sp.]	tRNA insert ORF 62	91	9	Major capsid protein	Hypothetical protein
3	34,14	Phage Major Capsid Protein [*Bacillus pumilus*]	PBSX-like ORF 11	95	10	Capsid protein	Phage Major Capsid Protein
4	16,45	Phage Portal Protein [*Bacillus* sp.]	PBSX-like ORF 18	115	12	phage portal protein	Phage tail tube protein

**Table 5 ijms-22-08164-t005:** Number of phages, with the available genome at NCBI *, described using a *Bacillus* strain as host (in alphabetical order).

*Bacillus* Specie	No. Phages	*Bacillus* Specie	No. Phages
*B. alcalophilus*	1	***B. pumilus***	**14**
*B. anthracis*	3	*B. safensis*	1
*B. bogoriensis*	1	*B. subtilis*	21
*B. cereus*	31	*B. thuringiensis*	44
*B. clarkii*	1	*B. velezensis*	1
*B. halmapalus*	1	*B. weihenstephanensis*	1
*B. licheniformis*	1	*Bacillus* sp.	7
*B. megaterium*	17		

* NCBI: https://www.ncbi.nlm.nih.gov/genome/viruses/; last access 1 May 2021.

**Table 6 ijms-22-08164-t006:** *B. pumilus* phages found in viral genome database at NCBI *. The name of the phage, the strain used for multiplication, the access number and genome size are detailed.

Phage Name	*B. pumilus*(B. p) Strain	Acc. Number	Genome Size
Bobb	*B. p* SAFR32	NC_024792	160,281
Bp8p-C	*B. p* GR8	NC_029121	151,417
Bp8p-T	*B. p* GR8	NC_047744	151,419
phiGATE	*B. p* GL1	NC_020081	149,844
Bpu_PumA	*B. p*	NC_049971	18,466
Bpu_PumB	*B. p*	NC_049972	18,932
Andromeda	*B. p* BL8	NC_020478	49,259
Blastoid	*B. p* BL8	NC_022773	50,354
Curly	*B. p* BL8	NC_020479	49,425
Eoghan	*B. p* BL8	NC_020477	49,458
Finn	*B. p* BL8	NC_020480	50,161
Glittering	*B. p* BL8	NC_022766	49,246
Riggi	*B. p* BL8	NC_022765	49,836
Taylor	*B. p* BL8	NC_041858	49,492

* NCBI: https://www.ncbi.nlm.nih.gov/genome/viruses/; last access 1 May 2021.

**Table 7 ijms-22-08164-t007:** Activity of Bp15.1PLP on bacterial strains from a laboratory collection.

Strain/Group	Lysis Activity	Inhibition Halo Morphology
*B. pumilus* 15.1	−	−
*B. pumilus* 15.1C	+++	Translucent
*B. pumilus* 8A3	+++	Translucent
*B. pumilus* M1	+++	Translucent
*B. pumilus* M2	+++	Translucent
*B. subtilis* sp.	−	−
*B. subtilis* 168C	−	−
*Bacillus thuringiensis* 78/11	−	−
*Bacillus cereus* 169	−	−
*Latococcus garvaeae*	−	−
*Lactococcus* NZ9000	−	−
*Escherichia coli* DH5α	−	−
*Arthrobacter* sp.	−	−
*Listeria innocua* CECT 4030	−	−
*Micrococcus* sp.	−	−

(−) indicates no lysis activity; (+++) indicates strong lysis activity.

**Table 8 ijms-22-08164-t008:** Mortality results, obtained after 10 days in *C. capitata* larvae bioassays using sporulated cultures of *B. pumilus* 15.1 and different concentrations of Bp15.1PLPs. The increase in toxicity compared to the negative control (water) was also calculated (Fold increase column).

Bioassay	% Mortality	Fold Increase
H_2_O	6	1.0
SM buffer	12	2.0
Sporulated culture	20	3.3
Bp15.1PLP (Lysate)	0	0.0
Sporulated culture + Bp15.1PLP (Lysate)	14	2.3
Bp15.1PLP (200x concentrated)	12	2.0

## Data Availability

All the data presented in this study are insert in the article or in the Appendix A.

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
