# Peer review of "Bacillus pumilus* 15.1, a Strain Active against *Ceratitis capitata*, Contains a Novel Phage and a Phage-Related Particle with Bacteriocin Activity"

_ijms, 2021, doi:10.3390/ijms22158164_

Round 1

Reviewer 1 Report

Reviewing this paper was particularly uneasy because of lack of page and line numbers.

This manuscript describes a region of B. pumillus encoding one PBSX-like region and a genomic island containing all or part of a bacteriophage. The 98kb region identified by PHASTER as encoding phage genes inside a 113kb contig (Contig 59) also contains bacterial genes and this makes understanding the results quite difficult. The PBSX-like region is 28.7kb long, while the region flanked by two 45bp repeats belonging to a tRNA gene is 60.2kband and encodes many genes that may not be phage-related. Figure 1 is said to show an organization of the 98kb region. It displays a group of bacterial genes in black followed by the PBSX-like region of 28,7kb and a supposedly 60,2kb region, marked 31,5kb on the figure. The 28.7kb subregion 2 bearing phage structural genes is not visible on Figure 1. This may be an editing problem but it did not help in reviewing the manuscript.

The authors should explain on which basis they have separated the 60.2kb region into two subregions, although they show a similar GC-content. Are the genes involved in DNA replication (subregion 1) of phage or bacterial origin? They should also be more clear about the capacity of this region to produce the siphovirus shown in Figure 10. Are all the necessary genes present? It is stated in the concluding remarks that phage Bp15.1Hope is encoded by the 60.2kb region but there is no actual proof of this, and this statement should be deleted. The siphoviruses must be separated from Bp15.1PLP which seems to be possible with a sucrose gradient, and their genome analysed before concluding about their origin.

There are multiple English errors that should be corrected. I have indicated a few but there are many more throughout the text.

Specific points

Abstract

Correct the following English errors

Genes « codifying for »

Was comprised by

Origin

Introduction

Correct the sentence: “The crystal… is mainly comprised by the enzyme”

The sentence “With aim of searching…This region was studied in detail” should be at the end of the introduction

Material and Methods

Pellet instead of precipitate after ultracentrifugation

HindIII not HinDIII

Results

Paragraphe 3.4: it is not clear: what is the meaning of "(putative insertion)" in the title? Does it mean “putative insertion site”?

Paragraphe 3.8

Protein analysis show major structural proteins and minor proteins. Protein 2 which appear to be minor is in fact a constituent of the second phage (Bp15.1Hope) contaminating the BP15.1PLP preparation. Therefore it cannot be stated that BP15.1PLP possesses these two capsid proteins.

The following sentence does not make any sense: “Phages found in B. pumilus (compiled in Table 6) showed a genome size ranging from 18,466 nt (phage Bpu_PumA1) to 160,281 nt (phage Bobb from B. pumilus SAFR32), with an average genome size (n=14) of 74,828 nt, similar to the 60.2 Kb inserted fragment found in B. pumilus 15.1.” What does “average genome size” mean? There is no such thing as an average phage.

Reviewer 2 Report

In this study, the authors report a specific DNA region in the entomopathogenic strain B. pumilus 15.1 encoding a 28.7 kb PBSX-like particle and novel phage, named Bp15.1PLP and Bp15.1Hope, respectively. Specifically, the authors performed a great amount of the bioinformatic analysis on the two regions. When B. pumilus was induced with mitomycin C, two phage particles were observed. Bp15.1PLP has bacteriocin activity against some of B. pumilus strains and no toxicity on C. capitata larvae but the Bp15.1Hope is under study.

Major comments

  1. The two types of phage-particles were predicted in B. pumilus 15.1 genome. Have the authors been able to separate them or confirm their DNA sequence after purification?
  2. In the section 3.3, ANI between 28.7 Kb and PBSX/PBSZ is more than 60% but how do the 28.7 Kb and PBSX/PBSZ have no nucleotide similarity?
  3. In the section 3.8, the authors mentioned a putative novel phage from the 60.2 Kb region but it would be good if it is moved into the new section for the observation.
  4. If the authors found two different types of phage, were they able to see two different band after sucrose gradient ultracentrifugation?
  5. In the bacteriocin assay, have the authors considered the effect of mitomycin C itself to B. pulmilus? It is not mentioned in the materials and methods whether the authors used phage lysate or purified phage stock.
  6. In the section 3.10, the authors said that Bp15.1PLP preparation showed no toxicity even when it was 200 times concentrated, but it shows 12 % mortality in the table.
  7. It seems that the authors were not able to separate the two phages, so I concern that all the data, except for bioinformatic data, is the results from the mixed phages. Then the subtitles that mention only Bp15.1PLP may confuse readers.

Minor comments

  1. Please stick to 98.1 Kb as genome size. In abstract, A 98Kb --> 98.1Kb
  2. In abstract, ‘…although at a low concentration.’ is not necessary.
  3. PBSX --> PBSX-like
  4. In the section 2.6, HinDIII --> HindIII
  5. In the section 2.6, please rewrite about digestion with restriction enzymes and ligation.
  6. Phage-like particles mean both PBSX-like particle and novel phage? At some point, it is confusing which one the authors describe between PBSX-like particle and novel phage.
  7. In the section 3.2, what percentage identity is 28.7 Kb region with PBSX from B. subtilis 168?
  8. In the section 3.4, it is confusing that ‘28.7 Kb’ is used in both as subregion2 within 60.2Kb and as PBSX-like particle.
  9. In the section 3.4, CG content --> GC content
  10. Table 7. Please specify what ‘-‘ and ‘+++’ means.

11. What does the two numbers in Fold increase mean in the Table 8?

Round 2

Reviewer 1 Report

The majority of this reviewer's request have been taken into account. The revised version is satisfactory.